# TeamMate: A Longitudinal Study of New Zealand Working Farm Dogs. III. Factors Affecting the Risk of Dogs Being Lost from the Workforce

**DOI:** 10.3390/ani11061602

**Published:** 2021-05-29

**Authors:** Katja E. Isaksen, Lori Linney, Helen Williamson, Elizabeth J. Norman, Nick J. Cave, Naomi Cogger

**Affiliations:** 1School of Veterinary Science, Massey University, Palmerston North 4410, New Zealand; n.j.cave@massey.ac.nz (N.J.C.); n.cogger@massey.ac.nz (N.C.); 2Vetlife, Timaru 7910, New Zealand; lori.linney@vetlife.co.nz (L.L.); helen.williamson@vetlife.co.nz (H.W.); 3College of Sciences, Massey University, Palmerston North 4410, New Zealand; e.j.norman@massey.ac.nz

**Keywords:** risk factors, longevity, death, euthanasia, retirement, longitudinal, TeamMate, working dogs, herding dogs, working farm dogs

## Abstract

**Simple Summary:**

Working farm dogs are essential to many livestock farmers, but little is known about what increases their risk of being lost from the workforce through dying, being euthanized or being retired. A study carried out in New Zealand found that the majority of farm dogs that were lost from work during a four-year period died or were euthanized rather than being retired, and owners reported that acute injuries or illnesses were the most common cause. However, 65% of dogs that died or were retired were at least seven and 38% at least 10 years old, showing that working farm dogs often work into their old age. Data from physical examinations performed by veterinarians showed that lameness almost doubled dogs’ risk of being lost from work, independently of their age. Our results show that further research into what causes lameness in working farm dogs, and how this lameness can be avoided, could make a significant positive impact on the health and welfare of these dogs.

**Abstract:**

Working farm dogs are essential to many livestock farmers. Little is known about factors that influence dogs’ risk of being lost from work. This paper explores risk factors for farm dogs being lost through death, euthanasia and retirement. All enrolled dogs were working and a minimum of 18 months old. Five data collection rounds were performed over four years. Data about dogs were collected from owners and dogs were given physical examinations by veterinarians. Dogs that were lost from work were counted and owner-reported reasons for loss were recorded. Multivariable logistic regression modelling was used to investigate risk factors for loss. Of 589 dogs, 81 were lost from work. Of these, 59 dogs died or were euthanized and 22 were retired. Farm dogs tended to reach advanced ages, with 38% being 10 years or older when last examined. Acute injury or illness was the most commonly owner-reported reason for loss. Age group (*p* < 0.0001) and lameness (*p* = 0.04, OR = 1.8) significantly affected dogs’ risk of being lost. These results expand our knowledge about factors that affect health, welfare and work in farm dogs. Further investigation into reasons for lameness may help improve health and welfare in working farm dogs.

## 1. Introduction

A range of concerns exist around the health and welfare of working farm dogs. Previous studies have found that traumatic injuries and musculoskeletal conditions are common in farm dogs in New Zealand [1,2,3,4,5]. Further, owners report concern that as many as 19% of their dogs may be underweight [3]. Supporting this, approximately one-third of dogs enrolled in the TeamMate study could be considered as underweight if assessed using body condition scoring [4]. However, it is not known how specific conditions or whether having low body condition scores affect the health or welfare of farm dogs. 

In addition to affecting welfare, it is plausible that factors related to health affect the risk of farm dogs being lost from the working population. Working farm dogs are an essential part of livestock farming in New Zealand and in other parts of the world [6,7]. The loss of dogs from work can be disruptive to the effective running of farms and put extra pressure on farmers and their remaining dogs. Knowing which factors are likely to increase dogs’ risk of death or retirement can help dog owners and veterinarians mitigate those risks to ensure that dogs have the longest and healthiest working lives possible. Such knowledge may also help inform further research into how the identified risk factors might be avoided. For example, in addition to being commonly recorded in working farm dogs, musculoskeletal injury and disease have been reported as common causes of euthanasia and death in police, military and guide dogs [8,9,10]. However, while cross-sectional studies have been carried out into reported reasons for dogs being lost from work, studies that analyze longitudinal data to investigate which factors might put dogs at increased risk of death or retirement are rare. Such risk factor analysis can reveal exposures that make dogs more susceptible to developing the conditions that cause them to be removed from work. Due to this lack of investigation there may be important risk factors that are currently being overlooked by researchers and veterinarians. 

The aims of this study were to fill this gap in knowledge by investigating risk factors that influenced death, euthanasia or retirement of dogs during the course of the TeamMate study. Additionally, owner-reported reasons for death or retirement were reported and compared with the significant risk factors revealed by our analysis. Determining whether specific factors related to demographics, husbandry and health are associated with the risk of working farm dogs dying, being euthanized or being retired will help researchers, veterinarians and dog owners decide which areas to focus on to improve dogs’ care and husbandry.

## 2. Methods

### 2.1. Study Design

TeamMate is a longitudinal study focusing on working farm dogs on the South Island of New Zealand. The study design and data collection procedure are presented in detail in a previous publication [4]. To summarize, a total of 126 dog owners associated with 116 farms participated in this study and 641 working farm dogs were enrolled. All working farm dogs belonging to participating owners were included if they were least 18 months old and working with livestock regularly. 

Data collection began in May 2014. Data were collected approximately every eight to nine months subsequently, and data from five data collection rounds were included in the current study. The fifth data collection round was completed in November 2017. Figure 1 is a flowchart showing the start dates for each data collection round and how many dogs, owners, and farming properties were enrolled at each round. At each data collection round, farm dog owners were visited on the farm where they worked, new dog owners and dogs were enrolled, and data were collected. New dog owners and dogs were enrolled up to and including the third data collection round. New dogs included dogs belonging to previously enrolled owners that had been acquired or had become old enough to be enrolled in this study between farm visits. Some new properties were registered subsequently to the third data collection round due to participating dog owners moving or changing jobs.

At each farm visit, including on enrolment, all enrolled dogs were physically examined by veterinarians, and dog owners were interviewed to collect information about dogs’ husbandry, feeding, and work. Scribes were responsible for filling in the questionnaires and taking note of any clinical findings. The physical examination included manipulation of all the major limb joints and examining dogs to for lameness. Lameness examinations consisted of making dogs trot dogs on a lead for a short distance directly away from and towards the examining veterinarian to allow them to examine the dog’s gait. All physical abnormalities were recorded, irrespective of their clinical significance. The questionnaires that were used as part of this study are available as Supplementary Materials to a companion research article [4].

All veterinarians and scribes were trained to ensure data collection was performed in a standardized way, with veterinarians asked to record specific clinical signs rather than diagnoses. Training included a run-through of all questionnaires and how they should be completed, as well as practical sessions that involved filling in the questionnaires and examining, scoring, and measuring farm dogs. During training sessions, normal ranges of motion of the joints were demonstrated in healthy working farm dogs. 

Abnormalities noted on physical examination were systematically categorized based on the examining veterinarian’s clinical notes. Categories were not mutually exclusive, and dogs could have multiple recorded abnormalities, also in the same anatomical location. Categorization was carried out by a single veterinarian (LL). Categories that were considered unclear or incomplete during data entry were checked by a veterinarian (LL and/or NJC). The complete system used to categorize physical abnormalities is available as Supplementary Materials to a companion research article [4].

In the current study, we included all examinations of dogs where no data were missing from the relevant explanatory variables and information was available on whether dogs died or were retired subsequently to the examination. To avoid an excessive reduction in our sample size, potential explanatory variables were not included in the multivariable analysis if more than five percent of examinations did not have a recorded value for the variable. See Figure 2 for details on how many examinations were excluded and the reasons they were excluded. 

### 2.2. Outcomes—Absence, Death and Retirement of Dogs

The outcome variable analyzed in this study was whether or not dogs were lost from the workforce through death or retirement. At each farm visit following their enrolment in this study, dogs were classified as present or absent. Dogs that were still working on the property but not available for physical examination on the day of data collection were classified as being present but not examined. The fates of dogs that were present on the last farm visit made to the owner were recorded as ‘working with original owner’. Absent dogs were classified as having been lost from the workforce if the owner reported them as having died or having been retired from work for any reason. Absent dogs that were not dead or retired were not classified as having been lost from the workforce, and their fates were categorized as ‘rehomed’, ‘sold’, ‘loaned’ or ‘withdrawn from the study’. Dogs reported as loaned included both dogs that had been loaned out to a different owner and dogs that had been returned to their owner after being loaned. Dogs were occasionally reported as having been retired to a smaller farm. These dogs were assumed to still be working, although in a reduced capacity, and were recorded as having been rehomed rather than being lost from the workforce. Where possible, the reason why a dog was absent was recorded. No data were available on whether health events or conditions that were reported by the owner as being the cause of a dog being absent, had been confirmed or diagnosed by a veterinarian. 

Data were analyzed to assess the risk of the dog dying or being retired following each farm visit and physical examination. Examinations where no information was available regarding the further fates of dogs were excluded from analysis. These examinations were either the last before a dog owner withdrew from this study or recorded during the final round of data collection (Figure 2). 

### 2.3. Explanatory Variables

Variables that were considered to be potential risk factors relating to dogs dying or being retired were screened for inclusion in the analysis. These variables are listed in Table 1. 

Clinical abnormalities were grouped according to their overall type—mainly whether the abnormality affected a specific body system—and were included in the analysis if they were present in 10% of dogs or more on enrolment in the TeamMate study [4]. Clinical abnormalities were analyzed as binary categorical variables, recording only presence or absence of each type of abnormality. While analyzing risk factors in this way meant losing some of the detail that was available in our data, we felt that the increased power of our models outweighed this disadvantage. Adding more levels to the clinical data might have caused us to miss higher level associations between clinical abnormalities and changes in the risk of death, euthanasia or retirement from work.

Lameness can be caused by musculoskeletal injury or disease, but also by other conditions such as footpad abrasions or nail injuries. Similarly, not all the clinical abnormalities that were included in the musculoskeletal category cause lameness. Therefore, despite often being associated with musculoskeletal conditions, lameness was analyzed as a separate risk factor to musculoskeletal conditions.

Examinations conducted on each dog were numbered, with enrolment being Examination 1, and each following examination being numbered sequentially. These examination numbers were included in the analysis to account for the progression of time during the course of this study.

Dog types were classified as ‘Heading dog’, ‘Huntaway’ or ‘other’ based on information provided by the dog owner. The types of work dogs were reported to carry out were classified as ‘heading’, ‘hunting’, ‘yard work’ or combinations of these. More details on the different types of working farm dogs found in New Zealand, how dogs enrolled in TeamMate were classified, their average body weights, which types of work they were recorded to do, and how these types of work were classified can be found in an earlier publication [4].

Body condition was scored using a validated nine-point numeric scale, where 1 is severely underweight, 9 is morbidly obese, and 4 to 5 is considered ideal [11]. Body condition in relation to dogs’ lean body mass was quantified by calculating the ratio of the predicted lean mass to skeletal size using a novel equation developed by measuring lean mass and body size in 20 working farm dogs [12].

Dog owners were asked to report the ages of dogs on the enrolment of dogs and on follow-up examinations. At 10% of examinations subsequent to enrolment, dogs’ ages were not recorded. In these cases, the dog’s age was calculated based on the dog’s reported age at enrolment and the time passed.

Dogs that were enrolled in TeamMate and had information available relating to whether or not they had died or been retired following at least one examination were eligible to be included in the current study. Data from examinations were excluded from multivariable analysis if they contained missing values in any of the variables that were examined as potential risk factors.

### 2.4. Statistical Analysis

The number of enrolled dogs were counted stratified by their fate at the conclusion of this study. Dogs that died or were retired were counted and stratified by the reported reason for death or retirement and their age group on their last examination in this study. Percentages and 95% CIs were calculated for all stratified counts.

The risk of a dog dying, being euthanized or being retired after each farm visit was analyzed using univariable and multivariable binary logistic regression models. Odds ratios were calculated by exponentiating the model β-coefficients. All models were checked for significance using *p*-values derived from log-likelihood ratio tests (LRT). Potential risk factors were included in the multivariable analysis if they had a recorded value in at least 95% of dog examinations, and the significance of the log-likelihood ratio test was less than *p* < 0.2 during univariable screening. The best-fit multivariable model was developed using backwards single-term deletion, where all potential risk factors were included in the first model tested and the variable with the smallest association with the risk of death or retirement was removed at each step. Backward elimination continued until all variables had a *p*-value for the log-likelihood ratio test of less than 0.05, which was considered to be statistically significant. The examination numbers for each dog were retained in all multivariable models, irrespective of effect size or significance, in order to account for the passing of time from the first to the last data collection round. Pairwise interactions were tested for all variables in the final multivariable model. To account for repeated measures over the course of this study, individual dog and dog owner identification numbers were added to the final multivariable model as nested random effects. The change in model fit caused by adding the random effects was tested using a log-likelihood ratio test.

All continuous explanatory variables were checked for linearity. The log-odds probabilities of dogs being lost were plotted against each continuous variable using a smoothed (loess) line [13]. The resulting plot was visually examined for linearity. Additionally, a quadratic term was added to the univariable model to allow the regression line to follow a curved path [13]. The quadratic term was created by centering and squaring the values of the variable. Centering was performed to avoid collinearity with the original predictor. The assumption of linearity was checked by examining whether the quadratic term was significantly associated with whether dogs died or were retired. If the *p*-value extracted from a log-likelihood ratio test was smaller than 0.05 and the smoothed line of the log-odds probabilities had a clear curvature, the assumption of linearity was determined to have been broken. In such cases, the explanatory variable was converted to a categorical, removing the assumption of linearity from the model.

To evaluate the quality of fit of the final multivariable model, we examined the area under the receiver operator characteristic (ROC) curve. Additionally, the residuals generated by the mixed logistic regression model were checked for outliers that might indicate problems with model fit.

All calculations and data analysis were performed using R version 4.0.x [14]. The values necessary to plot loess smoothed lines for checking linearity of continuous predictors were generated using the loess() function in the car package [15] and the logit() function in the stats package [14]. Random effects models were fitted using the lme4 package [16]. The receiver-operator curve was generated and plotted using the pROC package [17]. The residuals of the final multivariable mixed model were plotted using the qqnorm() and qqline() functions in the stats package.

## 3. Results

In total, 1930 examinations were recorded from the 641 dogs that were enrolled in TeamMate. Four hundred and ninety-three examinations were removed from the dataset due to a lack of information about the fate of the dog following the relevant examination or due to missing data in variables that were examined as risk factors for death or retirement. Figure 2 shows how many examinations were removed from the final dataset and which explanatory variables contained missing values. Full sets of data with no missing examinations in the relevant variables were available for 1360 examinations of 589 working farm dogs belonging to 120 dog owners. Table 2 shows the distribution of dogs by sex, age group at enrolment and type.

### 3.1. Fates of Dogs and Reasons for Loss

In total, 81 of 589 dogs had the outcome of interest, that is they were lost from the workforce through dying, being euthanized or being retired during the study period. Table 3 lists the fates of all 589 dogs following the last examinations that were included in this study, Table 4 shows the owner-reported reasons why dogs died, were euthanized or were retired, and Table 5 shows the age groups of dogs that died or were retired.

### 3.2. Analysis of Risk Factors for Loss

Dogs’ ages, body weights, body condition scores and ratio of predicted lean mass to skeletal size were found to have non-linear relationships with the log-odds of dogs dying, being euthanized or being retired. These potential explanatory variables were therefore converted to categorical values before analysis. Dogs’ ages were categorized using the age groups used in a previous publication [4]. Dogs’ body weights were divided into quartiles. Body condition scores were categorized according to whether they are considered underweight (1–3), ideal (4–5) or overweight (6–9) according to the World Small Animal Veterinary Association [11]. The ratio of predicted lean mass to skeletal size was divided into quartiles.

Table 6 shows an overview of potential explanatory variables that were excluded from univariable screening due to having recorded values in less than 95% of the 1522 examinations where dogs had a known fate. If these four variables had been included in the multivariable analysis, the sample size would have been reduced by 662 examinations of 349 dogs.

Univariable screening for significance was performed using the final dataset of 1360 examinations of 589 dogs. Table 7 shows an overview of potential explanatory variables that were excluded after univariable screening due to being insufficiently associated with the risk of dogs dying, being euthanized or being retired from work (*p*(LRT) > 0.2). The results of univariable testing of potential explanatory variables that were included in the multivariable analysis and model building are listed in Table 8.

Table 9 presents the odds ratios calculated from the best-fit multivariable logistic mixed model, and the change in model fit when each of the remaining explanatory variables were removed. Dogs in the youngest and oldest age groups had the highest risk of dying, being euthanized or being retired, with dogs between three and 4.9 years having the lowest risk. Dogs were twice as likely to die or be retired if they were recorded as being lame on trot. Additionally, the presence of eye abnormalities had an effect on the risk of dogs dying, being retired or being euthanized (odds ratio = 1.9, 95% CI = 1.0–3.6, *p*(LRT) = 0.06) after accounting for age group and the presence of lameness. While this effect did not meet the threshold for significance and was removed from the final multivariable model, the effect of eye abnormalities had a relatively much stronger effect than any other tested variables is therefore mentioned here.

The final multivariable mixed model had an area under the ROC curve of 0.76 (95% CI = 0.71–0.82).

## 4. Discussion

This is the first time that risk factors related to death, euthanasia or retirement have been explored in working farm dogs. Being lame on trot almost doubled the risk of farm dogs being lost from the workforce, suggesting that preventing dogs from becoming lame could reduce their risk of being lost from the workforce significantly. Due to the physical requirements of the work farm dogs do, this increase in risk is to be expected, particularly if the lameness is long lasting and cannot be effectively treated. Lameness can be a sign of musculoskeletal pain and stiffness in dogs and conditions such as cranial cruciate ligament disease and joint dysplasia are common causes of lameness in dogs [18]. However, dogs can also be lame from other causes such as trauma to the footpads. Because we did not incorporate diagnoses in our data, we do not know what caused the lameness that was recorded in dogs enrolled in this study and musculoskeletal abnormalities were analyzed as a separate risk factor from lameness. When analyzed, the presence of musculoskeletal abnormalities was not found to have a significant effect on the risk of dogs being lost from the workforce despite the significant effect caused by lameness and the likelihood that many cases of lameness were caused by underlying musculoskeletal conditions. One reason for this apparent discrepancy may be the way we analyzed our data. In this study, we examined what effect risk factors had in the months immediately following each examination of a dog. As musculoskeletal disease often develops over long periods of time before it progresses to cause pain or lameness, it may be that our analysis considered too short periods of time to detect the effect of musculoskeletal abnormalities on working farm dogs. However, it is likely that musculoskeletal injury and disease is the underlying cause of many the recorded cases of lameness in this study. Research into what types of disorders commonly cause lameness in working farm dogs would be helpful. The results of such research may enable dog owners and veterinarians to treat these conditions more effectively and delay or prevent dogs from developing lameness that could cause them to be removed from work.

Another possible reason for the apparent discrepancy between the effects of lameness and musculoskeletal abnormalities on dogs being lost from work could also be that dog owners are more able to recognize lameness in their dogs than more subtle signs of musculoskeletal disease. Decisions about whether to euthanize or retire working farm dogs are made by their owners, presumably based on their needs on farm and their perception of the performance, health and welfare of their dogs. Many of the most common types of musculoskeletal abnormalities recorded for TeamMate, such as reduced range of joint motion, crepitus and joint pain [4,5], are likely to be difficult for dog owners to detect. Changes in movement, behavior and performance in working farm dogs are probably easier for dog owners to notice than subtle musculoskeletal changes, especially when they know their dogs intimately and rely on them to be able to work. As such, musculoskeletal abnormalities are unlikely to affect owners’ decisions on whether to remove dogs from work until they are serious enough to cause dogs to become lame. Lameness and related problems with joint stiffness or pain can cause dogs to have difficulties with, for example, jumping up and down from vehicles or across fences, and to have reduced working performance [19]. A study of military working dogs found that dogs were more likely to have signs of spinal disease if they were reported by their handlers to have developed problematic behaviors such as reluctance to jump up onto objects or vehicles, reluctance to perform work tasks or to have become aggressive or anxious [20]. If musculoskeletal abnormalities can be detected and treated early enough, the development of irreversible disease can sometimes be slowed down [21]. Providing farm dog owners with the necessary skills to detect musculoskeletal abnormalities before they progress to cause lameness could enable them to seek veterinary treatment early enough to prevent more serious injury or disease from developing. For example, farm dog owners could be trained in how to detect subtle changes in joint motion or signs of pain in their dogs and to recognize changes in behavior and performance that may indicate pain or discomfort. Helping farm dog owners to recognize early signs of musculoskeletal disease in their dogs could help them to make informed decisions around treatment, retirement and euthanasia.

Age group and the presence of lameness had the strongest effects on the risk of death, euthanasia or retirement in working farm dogs (Table 9). However, acute injury or illness was the most commonly reported reason for dogs being lost from the workforce (Table 4). The high proportion of dogs reported as being lost due to acute injuries or illnesses can be linked to our analysis on risk factors. Dogs that are lame or are suffering from age-related reduction in body function are probably less able to cope with the physical demands of their work, putting them at increased risk of serious acute injuries that can cause them to be retired or euthanized. For example, dogs that are old or lame may plausibly be less able jump over obstacles such as fence lines or avoid being hit by vehicles or stock. Additionally, young dogs have been shown to require veterinary treatment for acute injuries more often than older dogs [22], possibly due to their lower levels of training and higher excitability. Dog owners could counteract the increased risk due to youth or seniority by adjusting dogs’ training and workload according to age. Additionally, prevention and effective treatment of the underlying causes of lameness should be a priority for veterinarians and working dog owners. Doing so would not only improve dogs’ health and welfare but could additionally prevent dogs from having serious injuries that cause them to be lost from the workforce prematurely. However, there is currently no data on how dog owners currently work to counteract injury and disease in their dogs or whether there are specific areas where improvements to common practices could be made.

Our results indicate that the risk of working farm dogs being lost from the workforce did not increase markedly until they were ten years or older (Table 9). Nearly three-quarters of dogs that were lost from the workforce were seven years or older on their last examination and, of retired dogs, three-quarters were ten years or older. Depending on body size, dogs can generally be considered as being senior at six to seven years of age and geriatric at approximately nine to 11 years [23], meaning that a majority of working farm dogs that were lost from the workforce could be said to have reached old age. For comparison, only 40% of police working dogs in New Zealand worked until the standard retirement age of eight years, and the median age at which police dogs left the workforce was 6.6 years [24]. Guide dogs in the United Kingdom worked for a mean of 8.5 years, or until they were approximately 10 or 11 years old [8], and military working dogs had a mean age at death of ten years [10]. Considering the high activity levels and potential for traumatic injuries, it is reasonable to expect that working farm dogs have shorter working lives than other working dogs such as police, military and guide dogs that are normally bred and trained by institutions and closely followed up by veterinarians throughout their lives. However, our data indicate that working farm dogs have similar or possibly longer careers than such working dogs.

A possible reason for the long recorded working lifespans of working farm dogs is that it can be difficult to define what retirement means in these farm dogs. Anecdotally, instead of having a clear cut-off point in either age or health status where dogs are removed from work and moved into retirement, farm dog owners make the decision on whether and how much their dogs should work based on their own knowledge and experience. Often this means that instead of being retired, older dogs’ workloads are gradually reduced according to their working capacity and performance, the owners’ needs, and the composition of the owners’ teams of dogs. If the owner has younger dogs that can replace the old dog satisfactorily, old dogs’ workloads will probably be reduced. However, fully trained and experienced older farm dogs can be very valuable to farmers, and owners may be reluctant to retire them as long as they are still able to work. Additionally, working farm dogs are highly motivated to work with stock even as they grow older and some may not be suitable to keep as house pets. As such their owners may feel that their welfare would be impaired if they were not allowed to work. Anecdotally, there were cases in TeamMate where older dogs were noted as being allowed to ‘tag along’ for work, and dogs as old as 14 were enrolled in this study [4]. These dogs are still exposed to risk factors related to work and may be at higher risk of injury due to lower physical capacity caused by aging [23]. Including such semi-retired dogs in our study population may have caused us to underestimate the number of dogs that would be considered as retired by their owners and to overestimate the number of dogs that die or are euthanized while still an active part of the workforce. However, we felt that excluding semi-retired dogs would be difficult to do in practice due to the lack of a clear definition about what constitutes retirement. We therefore chose to define all dogs that were reported to be still working in any capacity as not retired.

Although our findings indicate that working farm dogs may have equally long working lives as police, military and guide dogs, these results should be interpreted with caution. Due to differences in data collection and the statistical methods used to analyze data, it is difficult to compare the studies on police, military and guide dogs directly with our results, and any apparent similarities or differences in results may be misleading. Additionally, police, military and guide dogs carry out work that can be crucial to the safety of humans, and as such any decrease in working performance is likely to cause them to be removed from work. In comparison, a working farm dog that is performing poorly due to age or injury may cause its owner to lose time and money, or cause injury to itself or other animals. However, since such farm dogs can still be useful and opportunities often exists for other dogs on the team to take over the most demanding work, or for dogs to be rehomed to less demanding farms, underperforming farm dogs may be less likely to be removed from the workforce than underperforming police, military or guide dogs. If working farm dogs continue working while their performance is lowered due to illness or injury, this may compromise their welfare and prevent them from fully recovering. However, to determine whether husbandry practices can be improved, more data are needed on the health and workloads of older working farm dogs, and how farm dog owners make decisions about when to remove dogs from stock work.

The effect of age on dogs’ risk of dying or being retired was not linear. Instead, the lowest risk was seen in dogs between three and 4.9 years old, rather than the youngest group of 1.5 to 2.9 year olds (Table 9). This is similar to the analysis performed by Sheard [3], which showed that dogs that were 2 years old or younger, or older than 7 years old, were the most likely to be reported as having died in a 12 month period. Sheard also found that partially trained dogs were more likely to die than those that were fully trained. Due to differences in behavior, fitness and training level, young dogs are likely to be exposed to somewhat different risk factors than older dogs, and they may be at higher risk of traumatic injury due to their lack of experience and higher levels of excitability. In Sweden, a study of insured pet dogs found that young dogs have a higher risk of receiving veterinary care due to traumatic injury than older dogs [22]. Additionally, young and partially trained dogs are sometimes unsuitable for stock work in general or incompatible with their current owner. In Australia, stock dog handlers reported that 20% of acquired dogs failed to become trained working dogs [25]. Of these, 89% were dismissed due to problems around temperament and training with more than half of dismissed dogs being reported to have a lack of natural working ability. While dogs that are simply incompatible with their owner can be rehomed or sold, finding new homes for farm dogs that are generally unsuitable for work is likely to be difficult due to their high energy levels and need for stimulation. It is therefore likely that a proportion of such dogs are euthanized rather than rehomed or sold.

Although not statistically significant, the observed effect of eye abnormalities on the risk of death, euthanasia or retirement in working farm dogs warrants further investigation. Reduced eyesight is a plausible cause of increased risk of acute injury that can lead to dogs leaving the workforce. However, it is likely that not all eye abnormalities recorded in this study were associated with reduced eyesight. For example, a majority of the recorded abnormalities on enrolment in this study were described as ‘lens opacity’ [4]. A common cause of lens opacity—especially in older dogs—is lens sclerosis, a condition that is not associated with a reduction in vision [26]. A possible alternative explanation for the effect of eye abnormalities on the risk of dogs being lost from work could be that dog owners are more likely to remove dogs with visible cloudiness in the eyes, irrespective of whether the dog’s vision is impaired. As mentioned, such cloudiness is more likely to occur in older dogs and may be a contributing factor to dogs being euthanized or retired due to ‘old age’. While our model did adjust for age group, we lost a good deal of detail in our age data by categorizing dogs’ ages, and we can therefore not rule out that the effect of eye abnormalities was confounded by dogs’ ages. Further investigation should be carried out to confirm or disprove whether there is a significant effect of eye abnormalities on the risk of working farm dogs dying, being euthanized or being retired.

Neither body condition score nor the ratio of predicted lean body mass to skeletal size score were found to be associated with the risk of dogs dying or being retired. This should not be interpreted as a lack of risk associated with being clinically under- or overweight. Instead, our evidence indicates that these representations of body condition are not predictive of whether dogs die or are retired. It may be that one or both measures are associated with welfare, working performance or other types of health outcomes in working farm dogs. However, in the current study there is little evidence to suggest that either measure is a reliable indicator of overall health in lean, athletic dogs. Given that concern has been expressed that working farm dogs may in general be too thin, more investigation into what constitutes optimal body condition in relation to health in working farm dogs may be warranted.

This study has a range of limitations, mostly related to missingness or a lack of detail in the data that were analyzed. Firstly, we did not examine the risk of dogs having certain types of abnormalities but instead limited analysis to abnormalities affecting certain body systems, plus lameness. While more detailed information was available, we felt that including these in the analysis would have reduced the power of our models and made it difficult to detect any impacts on dogs’ risk of being lost from work. Instead, we performed the analysis on higher-level variables with the aim of providing direction for future investigations into health and loss of working farm dogs. For example, such investigations can focus specifically on lameness to determine what types of underlying conditions are associated with the presence of lameness and whether some of these increase dogs’ risk of dying or being retired more than others.

A second limitation of this study concerns the quality and integrity of the collected data, especially in the data surrounding dogs’ work and husbandry. In some cases, such as the data on how dogs were fed, the collected data were not detailed enough to make a meaningful analysis. In other cases, the data contained too many missing values to be included in the multivariable analysis without a concerning drop in sample size and power. As a result, we were unable to analyze certain risk factors at all while other were excluded from multivariable analysis. Some of these risk factors, such as the number of days dogs worked in the previous week, may have had strong effects on the risk of dogs being lost from work and should be examined in future studies. These problems with detail and data integrity were mostly the result of attempting to record a large amount of information at once, which makes it difficult to collect all the detail that would be useful in analysis while still making it easy to collect data correctly and effectively. Future studies that focus on aspects of health or husbandry of working farm dogs could collect more detailed data that can be analyzed in more detail. For example, the impacts of diet, portion sizes and feeding intervals on farm dogs’ health and performance are topics that deserve more focused research.

Thirdly, the long intervals between examinations and data collection for TeamMate made it impossible to determine or make a good approximation of when dogs had died or been retired. As a result, we chose to analyze the data using logistic regression modelling rather than using survival modelling, which would have enabled us to use more data from dogs that were still present in the workforce at the end of this study. For the same reason, we did not calculate the incidence rates of dogs being lost from the workforce. Future longitudinal studies should use data that were collected at shorter intervals that we were able to do with TeamMate as this would enable the use of more powerful analysis that utilizes more of the collected data.

## 5. Conclusions

This study found that age group and lameness had significant effects on the risk of working farm dogs dying or being retired from work, and that a majority were working with stock into old age. Although more can doubtless be done to investigate and improve the health and welfare of working farm dogs while they are part of the workforce, there is probably little to be gained in attempting to extend dogs’ working careers. However, due to a lack of quality in our data, we did not examine factors which may affect the risk of developing disease or lameness, or carry out any assessment on the effects of husbandry practices such as feeding or the quality of housing. Future investigations should focus on these issues, as they are important in allowing dog owners to improve the overall health and welfare of their working farm dogs. Additionally, efforts might be made to investigate health and welfare in older working dogs, and how owners make decisions around whether to euthanize or retire dogs whose ability to work is declining. Such investigation has the potential to greatly improve the health and welfare of working farm dogs.

## Figures and Tables

**Figure 1 animals-11-01602-f001:**
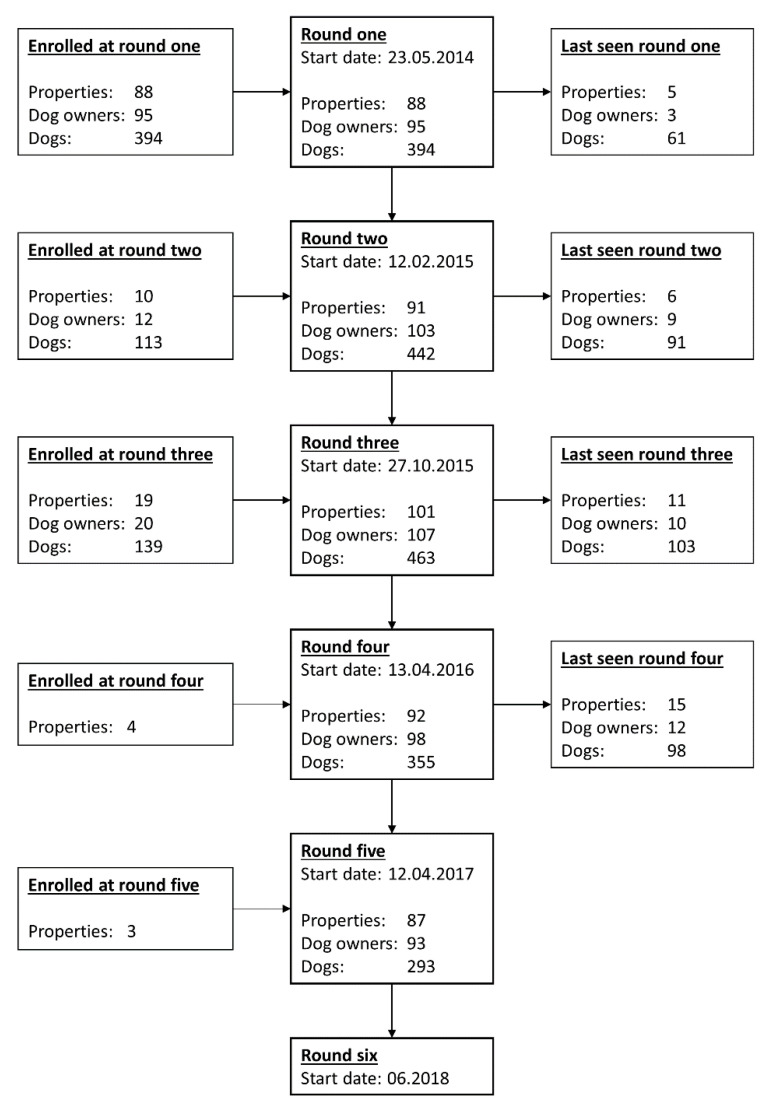
Flowchart showing the start dates of each data collection round as well as the number of farms, dog owners and dogs enrolled in TeamMate up to and including the fifth round of farm visits. Additionally, 14 properties, 16 dog owners and 68 dogs missed at least one round of data collection. Note that data for the sixth data collection round were not yet available at the time of writing. This figure was previously published by the authors [4] and is licensed for re-use under the Creative Commons Attribution 4.0 International licence.

**Figure 2 animals-11-01602-f002:**
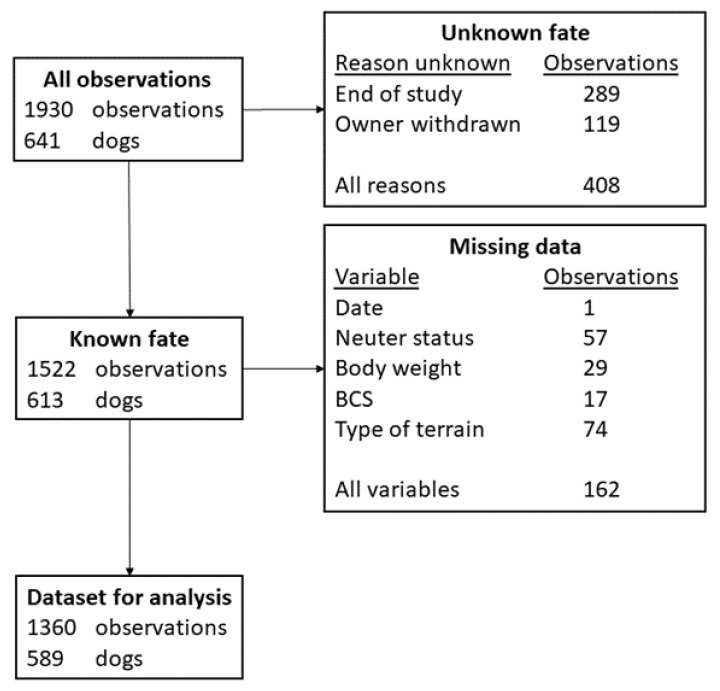
Flowchart showing the number of examinations that were removed from the analysis due to missing information. As examinations could have missing data in more than one variable, the sum of examinations with missing data in the different variables do not equal the total number of examinations that were removed. Note that dogs that had one or more examinations removed could still be present in the dataset.

**Table 1 animals-11-01602-t001:** List of explanatory variables that were assessed as possible risk factors for the death or retirement of working farm dogs.

Type	Variable Names
Examination	Number of examinations including enrolment.
Characteristics of dogs	Age, sex, neuter status, type of dog, body weight, body condition score (1–9), ratio of predicted lean body mass to skeletal size.
Findings on physical examination	Number of recorded abnormalities, presence of lameness on trot, presence of musculoskeletal abnormalities, presence of skin abnormalities, presence of mouth and teeth abnormalities, presence of eye abnormalities, presence of reproductive abnormalities.
Work related variables	Type of work, number of days worked in week preceding examination.
Other	Type of terrain on property, presence of bedding in kennel.

**Table 2 animals-11-01602-t002:** Population data relating to 589 working farm dogs that were enrolled in TeamMate and included in the risk factor analysis.

Variables		Number of Dogs	Percentage
Sex	Female	269	46%
	Male	320	54%
Age on enrolment	1.5 to 2.9 years	179	30%
	3 to 4.9 years	164	28%
	5 to 6.9 years	104	18%
	7 to 9.9 years	107	18%
	10 years and above	35	6%
Type of dog	Heading dog	282	48%
	Huntaway	288	49%
	Other	19	3%

**Table 3 animals-11-01602-t003:** The fates of 589 working farm dogs enrolled in TeamMate.

Fate of Dog	Number of Dogs	% (95% CI)	
Working with original owner	427	72	(69–76)
Dead or euthanized	59	10	(8–12)
Retired from work	22	4	(2–5)
Rehomed	32	5	(4–7)
Sold	44	7	(5–10)
Loaned	4	1	(0–1)
Not reported	1	0	(0–1)

**Table 4 animals-11-01602-t004:** Owner-reported reasons for death or retirement of 81 dogs enrolled in TeamMate.

	Died or Euthanized			Retired			All Dead or Retired		
	n = 59			n = 22			n = 81		
Reported Reason	Dogs	% (95% CI)		Dogs	% (95% CI)		Dogs	% (95% CI)	
Acute injury or illness	21	36	(23–48)	1	5	(0–13)	22	27	(17–37)
Old age	6	10	(2–18)	4	18	(2–34)	10	12	(5–20)
Chronic injury or illness	8	14	(5–22)	1	5	(0–13)	9	11	(4–18)
Sudden death	8	14	(5–22)	–	–	–	8	10	(3–16)
Behaviour	6	10	(2–18)	0	0		6	7	(2–13)
Not reported	10	17		16	73		26	32	

**Table 5 animals-11-01602-t005:** Age on last examination of 81 dogs enrolled in TeamMate that were reported as having died or been retired from work.

	Died or Euthanized			Retired			All Dead or Retired		
	n = 59			n = 22			n = 81		
Age on Last Examination	Dogs	% (95% CI)		Dogs	% (95% CI)		Dogs	% (95% CI)	
1.5 to 2.9 years	10	17	(7–27)	1	5	(0–13)	11	14	(6–21)
3 to 4.9 years	6	10	(2–18)	0	0		6	7	(2–13)
5 to 6.9 years	10	17	(7–27)	1	5	(0–13)	11	14	(5–22)
7 to 9.9 years	18	31	(19–42)	4	18	(2–34)	22	27	(17–37)
10 years and older	15	25	(14–37)	16	73	(54–91)	31	38	(28–49)

**Table 6 animals-11-01602-t006:** Number and percentage of examinations that had a recorded value for the respective variable. Odds ratios were calculated from the β-coefficients of univariable logistic regression models examining the association between potential explanatory variables and the risk of examinations being followed by dogs dying or being retired. *p*-values were derived from log-likelihood ratio tests of the same models. The listed explanatory variables were not included in multivariable analysis as they had recorded values in less than 95% of examinations. Data are from 1522 examinations of 613 dogs that were enrolled in the TeamMate project.

		Examinations				
Variable Name	Variable Levels	n	%	Died or Retired	Odds Ratio (95% CI)	*p*(LRT)
Number of days worked week before examination	(Count value)	1421	93	99	0.8 (0.8–0.9)	<0.001
Ratio of predicted lean body mass to skeletal size (quartiles)	Below 3.4	244	69	21	Ref	0.47
	3.5 to 3.7	257		13	0.6 (0.3–1.2)	
	3.8 to 4.1	271		19	0.8 (0.4–1.5)	
	4.2 and higher	272		19	0.8 (0.4–1.5)	
Presence of bedding in kennel	No	735	90	56	Ref	0.47
	Yes	636		42	0.9 (0.6–1.3)	
Types of work	Heading only	571	94	42	Ref	0.35
	Hunting only	59		7	1.1 (0.5–2.4)	
	Heading and yard work	102		8	1.7 (0.7–4.0)	
	Hunting and yard work	401		27	0.9 (0.6–1.5)	
	Heading, hunting and yard work	235		16	0.9 (0.5–1.7)	
	Other combinations	58		1	0.2 (0.0–1.6)	

**Table 7 animals-11-01602-t007:** The results of univariable screening of potential explanatory variables for the risk of examinations being followed by dogs dying, being euthanized or being retired. The *p*-values derived from log-likelihood ratio tests of univariable logistic regression models examining the association between potential explanatory variables and the risk of examinations being followed by dogs dying or being retired. The listed explanatory variables were not included in multivariable analysis as they had *p*-values larger than 0.2. Data are from 1360 examinations of 589 dogs, of which 81 examinations were followed by a dog dying or being retired. All dogs were enrolled in the TeamMate project and all examinations had recorded values for all tested variables.

		Number of Examinations			
Variable Name	Variable Levels	Working	Died or Retired	Odds Ratio (95% CI)	*p*(LRT)
Body condition score	1 to 3	453	35	Ref	0.21
	4 to 5	953	61	0.8 (0.5–1.3)	
	6 to 9	99	11	1.6 (0.7–3.6)	
Skin abnormalities	No	691	47	Ref	0.48
	Yes	588	34	0.9 (0.5–1.3)	
Sex	Female	590	40	Ref	0.57
	Male	689	41	0.9 (0.6–1.4)	
Body weight (quartiles)	21 kg and below	349	22	Ref	0.68
	21.1 to 25 kg	311	21	1.1 (0.6–2.0)	
	25.1 to 30 kg	338	17	0.8 (0.4–1.5)	
	30.1 kg and above	281	21	1.2 (0.6–2.2)	
Types of terrain	Flat and steep	694	47	Ref	0.78
	Flat	340	19	0.8 (0.5–1.4)	
	Steep	245	15	0.9 (0.5–1.6)	

**Table 8 animals-11-01602-t008:** The results of univariable logistic regression models examining the risk of each visit being followed by dogs dying or being retired in relation to a range of explanatory variables. β-coefficients (with standard errors (SE)) and odds ratios (with 95% CIs) derived from the logistic regression models and *p*-values derived from log-likelihood ratio tests. Explanatory variables with *p* < 0.2 are reported. Data are from 1360 examinations of 589 dogs, of which 81 examinations were followed by a dog dying or being retired. All dogs were enrolled in the TeamMate project and all examinations had recorded values for all tested variables.

		Number (%) of Examinations								
Explanatory Variables	Level	Working		Died or Retired		β-coefficient (SE)		Odds Ratio (95% CI)		*p*(LRT)
Age category	1.5 to 2.9 years	275	(20)	11	(1)	Ref		Ref		<0.0001
	3 to 4.9 years	402	(30)	6	(0)	−1.0	(−1.5–-0.5)	0.4	(0.1–1.0)	
	5 to 6.9 years	260	(19)	11	(1)	0.1	(−0.4–0.5)	1.1	(0.5–2.5)	
	7 to 9.9 years	265	(19)	22	(2)	0.7	(0.4–1.1)	2.1	(1.0–4.4)	
	10 years and older	77	(6)	31	(2)	2.3	(1.9–2.7)	10.1	(4.8–20.9)	
Number of recorded abnormalities	(count)	-		-		0.2	(0.1–0.2)	1.2	(1.1–1.3)	<0.0001
Eye abnormalities	No	1188	(87)	62	(5)	Ref		Ref		<0.0001
	Yes	91	(7)	19	(1)	1.4	(1.1–1.7)	4.0	(2.3–7.0)	
Mouth and teeth abnormalities	No	764	(56)	32	(2)	Ref		Ref		0.0004
	Yes	515	(38)	49	(4)	0.8	(0.6–1.1)	2.3	(1.4–3.6)	
Lameness on trot	No	1123	(83)	59	(4)	Ref		Ref		0.0005
	Yes	156	(11)	22	(2)	1.0	(0.7–1.3)	2.7	(1.6–4.5)	
Reproductive system abnormalities	No	1194	(88)	68	(5)	Ref		Ref		0.005
	Yes	85	(6)	13	(1)	1.0	(0.7–1.3)	2.7	(1.4–5.1)	
Musculoskeletal abnormalities	No	674	(50)	28	(2)	Ref		Ref		0.001
	Yes	60	(44)	53	(4)	0.7	(0.5–1.0)	2.1	(1.3–3.4)	
Neuter status	Entire	1189	(87)	68	(5)	Ref		Ref		0.01
	Neutered	90	(7)	13	(1)	0.9	(0.6–1.2)	2.5	(1.3–4.7)	
Visit number	(count)	-		-		0.3	(0.2–0.4)	1.3	(1.1–1.6)	0.01

**Table 9 animals-11-01602-t009:** Results of the final multivariable logistic mixed model showing the effect of a range of explanatory variables on the risk of examinations being followed by dogs’ dying or being retired. Individual dogs and dog owners were defined as nested random effects. Data used in the final model are from 1360 examinations of 589 dogs, of which 81 examinations were followed by the dog dying, being euthanized or being retired. All dogs were enrolled in the TeamMate project.

Explanatory Variables	Level	Odds Ratio (95% CI)		*p*(LRT)
Examination number	(count)	1.2	(0.9–1.5)	0.15
Age category	1.5 to 2.9 years	Ref		<0.0001
	3 to 4.9 years	0.3	(0.1–0.9)	
	5 to 6.9 years	1.0	(0.5–2.4)	
	7 to 9.9 years	1.8	(0.8–3.8)	
	10 years and older	8.3	(3.9–17.7)	
Lameness on trot	No	Ref		0.03
	Yes	2.0	(1.2–3.4)	
Dogs and dog owners (random effects)				1.0

## Data Availability

The datasets generated for this study are available on request to the corresponding author.

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
