# Peer review of "TeamMate: A Longitudinal Study of New Zealand Working Farm Dogs. III. Factors Affecting the Risk of Dogs Being Lost from the Workforce"

_animals, 2021, doi:10.3390/ani11061602_

Round 1

Reviewer 1 Report

The authors aims to fill the gap in knowledge of risk factors that influenced death, euthanasia or retirement of dogs during the course of the TeamMate study. Additionally, the relationship of owner-reported reasons for death or retirement with the significant risk factors revealed by the analysis. 

I reaffirm my gratitude for being able to review this manuscript. It is not easy to face complex datasets, with longitudinal data, with the risk of missing values and still carry out more advanced statistical techniques contributing to the generation of new knowledge, also considering that the work is based in small animals.

The authors have amended satisfactorily all my recommendations and deeply work with other reviewers suggestions, considering that the manuscript is in a condition to be published in its current form.

Reviewer 2 Report

Thank you for your responses. No further comments

Reviewer 3 Report

I appreciated the authors' attempt to revised the Manuscript according to the reviewers comments, even if some  critical issues  are still present. Anyway, I think  that  in this last round the new version could be taken into consideration for publication.

This manuscript is a resubmission of an earlier submission. The following is a list of the peer review reports and author responses from that submission.

Round 1

Reviewer 1 Report

This is a well written and well designed study. The results are interesting and important.

Minor comments:

Line 29 - consider advanced or old age rather than high age

Line 94 - because of the importance of the lameness in the final model, it would be helpful to have more detail on the assessment of lameness. The dogs were encouraged to trot - was it recorded whether they did trot or not? How was lameness scored - yes/no vs on a lameness scale? Was the exam video recorded? If it was simply yes/no - then the limitations of that type of assessment should be included in the discussion.

Although the data regarding work in the past week was unable to be analyzed, was there an attempt to document the season in which the exam was performed? was it in peak working season or in a low season? This may lead to a bias in the detection of lameness.

Line 203 and Table 6 - why was the 95% cutoff chosen? did you run the analysis using an alternate cutoff? it seems like there was still a lot of data available for some of the variables and the number of days worked may have impacted the lameness variable.

The limitations of the study/interpretation should be clearly delineated.

Reviewer 2 Report

Authors aims to fill the gap in knowledge of risk factors that influenced death, euthanasia or retirement of dogs during the course of the TeamMate study. Additionally, the relationship of owner-reported reasons for death or retirement with the significant risk factors revealed by the analysis. 

Small comments/suggestions highlighted in the attached .pdf file

The possibility of reviewing a manuscript using more complex data analysis methods is appreciated, especially in small animals. 

Perhaps the only thing that concerns mi, is that at some point the M&M section is hard to follow, due to the amount of information, if possible some further efforts on ordering this section, perhaps that could help the readers to follow-up throughout the manuscript.

Reviewer 3 Report

 TEAMMATE: A LONGITUDINAL STUDY OF HEALTH IN WORKING FARM 2 DOGS. III. FACTORS AFFECTING THE RISK OF DEATH, EUTHANASIA OR RE-3 TIREMENT FROM WORK.

The aim of this article was to investigate risk factors that influenced death, euthanasia or retirement of  farm dogs in New Zealand during the course of the Team-Mate study. Additionally, owner-reported reasons for death or retirement were reported and compared with the significant risk factors evaluated in the study.  

Areas of strength:

  • The topic could provide useful information to the literature

Areas of weakness:

  1. The reader is often forced to look for  data in articles previously published by the authors in different journals . This makes reading unpleasant and not very fluent
  2. Materials and methods section lacks detailed information about enrolled dogs (see below)
  3. Discussion must be improved on the basis of all the variables considered.

Lines 75-77

Signalment of enrolled dogs (type of dog, sex and age) must be presented in a table to improve clarity and comparisons for the reader

Lines 106-107

Can you please explain why categorization was carried out by a single veterinarian and cheked  by another person with training in animals health rather than by two veterinarians? The latter solution would be more compatible with a scientific method

Line 308

The sentence seems to be incomplete

Lines 457-475

You defined  variables that were assessed as possible risk factors for death or retirement in Table 1. It would be very interesting to discuss  also the effects of work related variables and  training level. In my opinion  these are important variables (not always related to the age of the dog), as could  be the type of work and the type of terrain dogs work on. Could you discuss more thoroughly these variables since you have a very large sample to work on and , if I understand correctly, you examined them in the questionnaire?

Lines 477-493

The sentences are questionable. Were the dogs retired only for eye abnormalities? Or eye abnormalities were associated with other diseases that could lead to retirement?
